# Characterization of a second class Ie ribonucleotide reductase
Juliane John[1], Daniel Lundin[1], Rui M. Branca [2], Rohit Kumar [1], Vivek Srinivas [1], Hugo Lebrette [3] ✉ & Martin Högbom [1] ✉

Class I ribonucleotide reductases (RNRs) convert ribonucleotides into deoxyribonucleotides under oxic conditions. The R2 subunit provides a radical required for catalysis conducted by the R1 subunit. In most R2s the radical is generated on a tyrosine via oxidation by an adjacent metal site. The discovery of a metal-free R2 defined the new RNR subclass Ie. In R2e, three of the otherwise strictly conserved metal-binding glutamates in the active site are substituted. Two variants have been found, VPK and QSK. To date, the VPK version has been the focus of biochemical characterization. Here we characterize a QSK variant of R2e. We analyse the organismal distribution of the two R2e versions and find dozens of organisms relying solely on the QSK RNR for deoxyribonucleotide production. We demonstrate that the R2e$_{QSK}$ of the human pathogen *Gardnerella vaginalis* (*Bifidobacterium vaginale*) modifies the active site-adjacent tyrosine to DOPA. The amount of modified protein is shown to be dependent on coexpression with the other proteins encoded in the RNR operon. The DOPA containing R2e$_{QSK}$ can support ribonucleotide reduction in vitro while the unmodified protein cannot. Finally, we determined the first structures of R2e$_{QSK}$ in the unmodified and DOPA states.

Ribonucleotide reductases (RNRs) are enzymes that produce deoxyribonucleotides, the building blocks of DNA, by reducing ribonucleotides. They can be divided into three classes according to their oxygen requirements: Class I is obligatory aerobic while class III cannot function in the presence of oxygen. Class II RNR is indifferent towards the presence of oxygen. Class I RNR consists of a larger catalytic subunit R1, which is allosterically regulated by different ribonucleotides and deoxyribonucleotides affecting substrate specificity and, in many cases, activity. Catalysis requires a radical that is generated by the smaller subunit R2 and subsequently shuttled to R1[1]. After substrate reduction, the radical is recovered and shuttled back to R2 where it is retained for subsequent use. Class I RNRs can be divided into subclasses a to e, mostly depending on the radical generating mechanism of R2[2]. R2b uses a dinuclear manganese site coordinated by four conserved carboxylate residues and two histidines. The manganese ions enable radical generation on a tyrosine close to the active site when exposed to a superoxide. This superoxide is produced by NrdI, a small accessory flavoprotein that binds to R2b and is oxidized by molecular oxygen[3]. The superoxide provided by NrdI is shuttled to R2b and generates a tyrosyl radical via oxidation of the metal site[4–6]. R2b requires NrdI for activity and the gene for NrdI, *nrdI*, is typically found near the gene for R2b,

*nrdF*, in genomes, as well as *nrdE*, the gene for R1. Another gene, *nrdH*, is also often found near *nrdF*. It encodes the small glutaredoxin-like protein NrdH. NrdH shows thioredoxin activity and can reduce class Ib R1 and the catalytic subunit of a subclass of the anaerobic class III RNR[7–9]. Independently of its role for RNR, NrdH has also been implicated in oxidative stress response[10,11] and sulphur assimilation[12].

The six metal ligands of R2b are highly conserved in *nrdF* genes[13,14]. However, already in 2008, Roca et al. found copies of class I R2 genes in which three of the metal-binding glutamates were substituted. They could demonstrate activity in vivo despite the substantially altered active site by complementing an RNR-deficient *Escherichia coli* strain with the full operon of the mutated RNR but did not see activity in vitro[6]. Recently, radical content and in vitro activity were demonstrated and crystal structures of R2e proteins were solved, including the radical-harbouring state[15]. The active protein had no metals bound and the radical harbouring tyrosine was *meta*-hydroxylated to a 3,4-dihydroxyphenylalanine (DOPA). The DOPA modification turned out to be mandatory for the activity of the RNR. This subclass of RNR was denoted class Ie[13,14]. R2e is phylogenetically derived from class Ib R2. There are two different versions of R2e known: The mutation to glutamine, serine and lysine (QSK) is the phylogenetically

[1]Department of Biochemistry and Biophysics, Stockholm University, Arrhenius Laboratories for Natural Sciences, Stockholm, Sweden. [2]Cancer Proteomics Mass Spectrometry, Department of Oncology-Pathology, Science for Life Laboratory, Karolinska Institutet, Solna, Sweden. [3]Laboratoire de Microbiologie et Génétique Moléculaires, Centre de Biologie Intégrative, CNRS – University of Toulouse, Toulouse, France. ✉e-mail: hugo.lebrette@univ-tlse3.fr; hogbom@dbb.su.se

ancestral variant while a derived clade has a mutation to valine, proline and lysine (VPK)[14]. The aforementioned publications demonstrated the activity of R2e$_{VPK}$ from three different organisms. In order to generate the DOPA modification in the R2e$_{VPK}$ protein, oxic coexpression with NrdI was necessary. When protein from this coexpression was purified, it was in the active, radical harbouring form. The mechanisms of DOPA formation and radical generation are presently unknown. NrdI likely has a similar function in R2e as in R2b by providing an oxidant, potentially protonated superoxide[16], that is shuttled to the active site via a channel in the R2-NrdI complex and used for radical generation. In short, different aspects of activity, functional data and structures have been demonstrated in many publications for R2b and, more recently, for R2e$_{VPK}$. For R2e$_{QSK}$ however—the R2 variant evolutionarily linking the two subclasses—no data is available. Srinivas et al. noted that there are a number of organisms, including important pathogens, that have the QSK variant RNR as their only class I RNR, which strongly suggests a functional RNR[14]. We were therefore interested in gaining a better understanding of R2e$_{QSK}$ and analysing similarities and differences compared to R2b and R2e$_{VPK}$.

Here we investigate the distribution of RNR class Ie and the gene neighborhood in R2e encoding organisms. Using X-ray crystallography and mass spectrometry we demonstrate that the QSK R2e variant from the human pathogen *Gardnerella vaginalis (Bifidobacterium vaginale)* generates a DOPA residue in the active site, analogous to what was observed for the VPK variant. Moreover, we show that the DOPA formation is facilitated by the coexpression of NrdI and unexpectedly also NrdH and R1 from the same organism. We tested the activity in vitro and found that the DOPA-containing protein is catalytically competent while the unmodified protein is not.

## Results
### Organism distribution of class Ie
The QSK variant of class Ie RNR is the evolutionary bridge between class Ib and the VPK variant of class Ie as has been shown by Srinivas et al.[14]. Since the genetic organization can be expected to evolve relatively slowly, we wanted to analyse the order and nature of genes near *nrdF* for the QSK and VPK subgroups. We searched the species representative genomes of the Genome Taxonomy Database (GTDB)[17] for *nrdF* genes and found 275 and 128 sequences containing *nrdF$_{QSK}$* and *nrdF$_{VPK}$*, in 266 and 128 genomes respectively (some genomes contain more than one gene for R2e$_{QSK}$). No genome was found to encode both R2e$_{QSK}$ and R2e$_{VPK}$. 132 species rely solely on subclass Ie RNRs for dNTP synthesis, 67 of which have *nrdF$_{QSK}$* and 65 *nrdF$_{VPK}$*. All R2e$_{QSK}$-encoding genomes were found in four orders of the *Actinomycetota* phylum (GTDB terminology), particularly the *Actinomycetales* (148) and *Mycobacteriales* (105) orders (Fig. 1, Supplementary Data 1). Similarly, the R2e$_{VPK}$ genes were mostly found in two *Bacillota* orders, *Mycoplasmatales* (93) and *Lactobacillales* (24). This distribution strongly suggests an influence of horizontal gene transfer of *nrdF$_{QSK}$* and *nrdF$_{VPK}$* genes between distantly related genomes, in line with what has previously been shown as a general pattern for RNR genes[18].

We could identify 65 species that have the QSK version as their sole RNR. The majority (40) of these are *Corynebacterium* spp., a *Mycobacteriaceae* genus containing species often encountered in host-associated microbiomes, sometimes found to be involved in disease[19].

Strikingly, in virtually all genomes encoding R2e$_{VPK}$, we found the genes for R1, R2 and NrdI in the order *nrdFIE* (Supplementary Table 1). Two gene stretches were missing the *nrdI* gene and only a single *nrdH* gene was found in these genomes (*nrdHFIE*). This gene, coding for the small NrdH thioredoxin known to co-occur with class Ib RNR genes, is, however, quite difficult to detect with certainty due to its short length and few conserved residues in its sequence. In the R2e$_{QSK}$ genomes, the order of genes was found to be more diverse and we did not find the R2e$_{VPK}$ order *nrdFIE* in any genome (Supplementary Table 1). We detected 17 of the *nrdHIEF* variant plus the variants *nrdIEF* (106) and *nrdIF* (104). In virtually all genomes that encode a gene stretch with an *nrdF$_{QSK}$* but lack an *nrdE*, another gene stretch containing an *nrdE*, sometimes with an *nrdF* (never

*nrdF$_{QSK}$*), sometimes without, could be identified (Supplementary Data 1). It hence appears that R2e$_{QSK}$ enzymes can work with R1 proteins that also work with normal dimanganese R2 enzymes. R2e$_{VPK}$ proteins that have been so far described in the literature stem from *Mesoplasma florum* (*Mycoplasmatales*), *Streptococcus pyogenes* (*Lactobacillales*) and *Aerococcus urinae* (*Lactobacillales*) (Fig. 1)[13,14].

The presence of over 60 organisms with the class I R2 QSK variant gene as their only RNR strongly suggests that the genes encode a functional RNR and makes it unlikely that they represent inactive gene remnants. The *Lactobacillales* order contains the species *Gardnerella vaginalis* (NCBI nomenclature), called *Bifidobacterium vaginale* in GTDB nomenclature, which is used in the text from here on. It encodes RNR$_{QSK}$ as its only class I RNR and also has genes for an anaerobic class III RNR. *B. vaginale* is a causative agent for bacterial vaginosis and mainly resides under anoxic conditions inside the human body, but it can be cultivated aerobically[20]. The genome of *B. vaginale* encodes all four proteins of the RNR$_{QSK}$ system near each other in the order *nrdHIEF*. We expressed R2e$_{QSK}$ from *B. vaginale* (*Bv*R2e$_{QSK}$) to spectroscopically and structurally investigate the properties of the QSK R2e variant.

### *Bv*R2e$_{QSK}$ expression under different conditions and spectroscopic characterization
Both R2b and R2e$_{VPK}$ require NrdI to produce a protein radical mandatory for RNR function. R2b can be recombinantly expressed as a metal-free protein and subsequently loaded with Mn(II) for successful metal cofactor reconstitution. Incubation with reduced NrdI and addition of O$_2$ leads to radical formation in vitro[5,21,22]. Contrariwise, similar protocols fail to produce an active R2e$_{VPK}$ and no experimental condition to generate the hydroxylated tyrosine in vitro has yet been described, although after DOPA formation during protein expression, the DOPA radical can be generated in vitro in R2e$_{VPK}$ via incubation with reduced NrdI and addition of molecular oxygen[14]. In order to produce both the tyrosine modification and a radical on the DOPA, R2e$_{VPK}$ has to be coexpressed with NrdI under aerobic conditions.

To investigate if the QSK variant of R2e behaves similarly to either R2b or R2e$_{VPK}$ proteins we designed different constructs of *Bv*R2e$_{QSK}$ and produced them in *E. coli* (Fig. 2A):
a) R2 alone: The *nrdF* gene with an N-terminal, cleavable his-tag for easy purification
b) R2 + NrdH: The same plasmid as for a) plus *nrdH* on a different plasmid
c) R2 + NrdI: *nrdFI* with a N-terminal, cleavable his-tag on R2
d) R2 + NrdI + NrdH: The same plasmid as for c) plus *nrdH* on a different plasmid
e) R2 + R1 + NrdI + NrdH: The operon *nrdHIEF* copied from the *B. vaginale* genome and cloned into a plasmid, plus a C-terminal uncleavable his-tag for easy purification.

Each construct was purified with metal ion affinity chromatography (IMAC) and size exclusion chromatography (SEC). *Bv*R2e$_{QSK}$ constructs carrying an N-terminal his-tag were subjected to TEV protease cleavage in order to remove the tag. The purification was conducted as fast as possible and finished about one and a half days after cell lysis. Unlike R2e$_{VPK}$ from *Mesoplasma florum* (*Mf*R2e) which is blue in the radical state and colourless in the non-activated or radical-quenched states[14], the *Bv*R2e$_{QSK}$ constructs were colourless when produced without NrdI and yellow when coexpressed with NrdI likely due to its binding to R2 (the oxidised FMN cofactor of NrdI is yellow). No colour was seen that would indicate an active radical. Ultraviolet-visible (UV-Vis) spectroscopy also did not indicate the presence of a radical (Fig. 2B), as no peak could be observed around 409 nm, which is typical for an RNR tyrosyl radical[4,23,24] or around 383 nm, diagnostic for the DOPA radical in R2e$_{VPK}$[13,14]. Therefore, the radical does not seem to be present in *Bv*R2e$_{QSK}$ after purification, regardless of the different combinations of proteins coexpressed. We also wanted to investigate if *Bv*R2$_{QSK}$ is lacking a metal cofactor, similar to R2e$_{VPK}$. To this end, we analysed the purified proteins by total reflection x-ray fluorescence (TXRF) spectroscopy

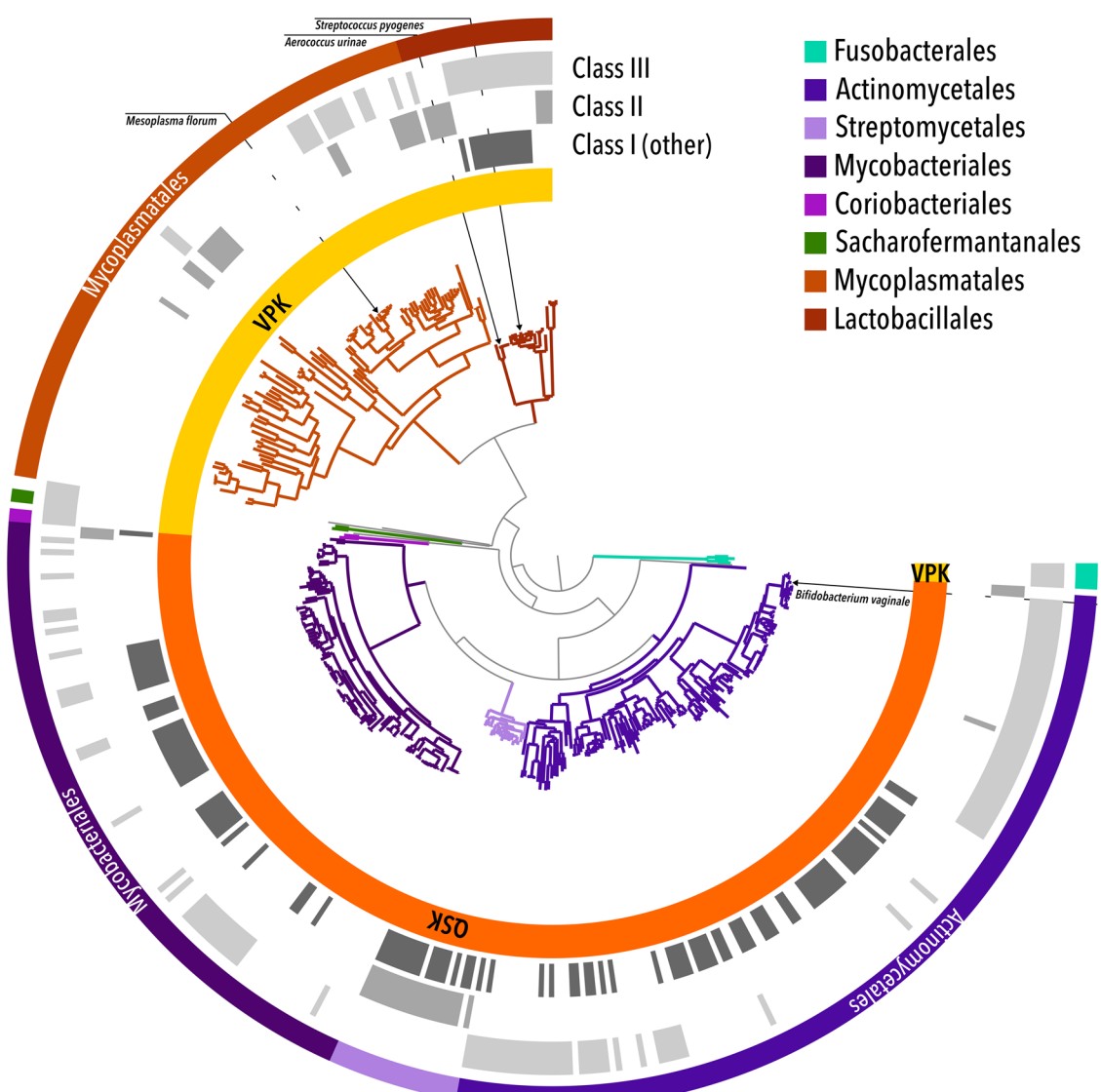

**Fig. 1 | Taxonomic distribution of R2e_QSK and R2e_VPK.** The middle of the figure shows a species tree of all species with R2e genes, using the GTDB nomenclature[17]. The innermost ring denotes if a genome encodes R2e_QSK or R2e_VPK. Rings 2-4 show if a genome encodes other RNRs. In the case of class I this means other class I than Ie.

The outermost ring, plus the colouring of the tree, shows the taxonomic orders of genomes. The four organisms containing class Ie RNRs investigated so far are marked.

in order to detect any metal present in the solution samples. Similar to the R2e_VPK variant, only trace amounts of manganese and iron were detected in the samples (Fig. 2C, Supplementary Table 2).

## Mass spectrometry of BvR2e_QSK expressed under different conditions

To investigate if the conserved tyrosine adjacent to the active site, residue 150 in *B. vaginale*, is modified similarly to the *meta*-hydroxylation of the tyrosine in R2e_VPK we conducted liquid chromatography-mass spectrometry (LC-MS) analysis of trypsin-digested protein produced from the different conditions of coexpression described above.

We analyzed all samples for peptides including position 150 on the protein. For the samples a) (R2 expressed alone) and b) (R2 coexpressed with NrdH) we could detect only peptides containing tyrosine. All samples with NrdI coexpression contained both peptides with tyrosine and DOPA – and no sample contained exclusively modified tyrosine (Fig. 3A).

We continued to compare the ratio of DOPA to tyrosine in the samples. In most cases, two different peptides do not have the same ionization efficiency in Electrospray Ionization Mass Spectrometry, and therefore comparing the respective peak intensities (as measured by integrating MS1

signal intensities over the retention time) does not produce a quantitative measurement. However, one can compare the peak intensities in a semi-quantitative manner to follow a trend across different samples, obtaining a rough estimation of the amounts of both species.

This comparison showed a surprising trend. First, as shown for R2e_VPK NrdI is required and sufficient for DOPA formation ($p$-value < 0.0001). More interestingly, the presence of NrdH enhances this formation further ($p$-value = 0.0041), but is not sufficient to modify the tyrosine alone. Strikingly, a clear additional increase of DOPA formation can be observed in BvR2e_QSK when the full operon is coexpressed ($p$-value = 0.0084), suggesting that R1 could also play a role (Fig. 3B). It seems unlikely that merely the gene neighbourhood and not the expression of R1 is responsible for the observed effect considering that the order of genes in the genome amongst R2e_QSK proteins is not well conserved. We note that in other class I RNR systems neither R1 nor NrdH have been suggested to influence R2 radical formation.

## In vitro activity of BvRNR_QSK

After observing the modification of the crucial active site tyrosine to a DOPA we wanted to examine the activity of the enzyme. To this end, the

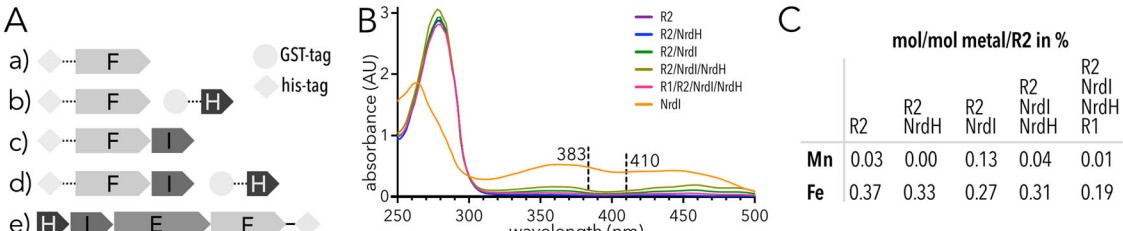

**Fig. 2 | Overview of the characteristics of *Bv*R2e$_{QSK}$ coexpressed with different combinations of proteins from the *nrd* operon. A** Summary of the constructs used to express R2 with different proteins from the operon. Tags are symbolized by a circle or a diamond; cleavable tags are shown connected with dotted lines, uncleavable with solid lines. **B** UV-Vis spectra of *Bv*R2e$_{QSK}$ expressed with five different conditions listed in **A**. NrdI is shown for comparison. Dashed lines at 383 nm and 409 nm correspond to R2a and R2e$_{VPK}$ radical peaks. Raw data for the graph can be found in Supplementary Data 1. **C** TXRF measurements of *Bv*R2e$_{QSK}$ produced according to the five conditions listed in **A**. The molar metal-to-protein ratio is given in percent.

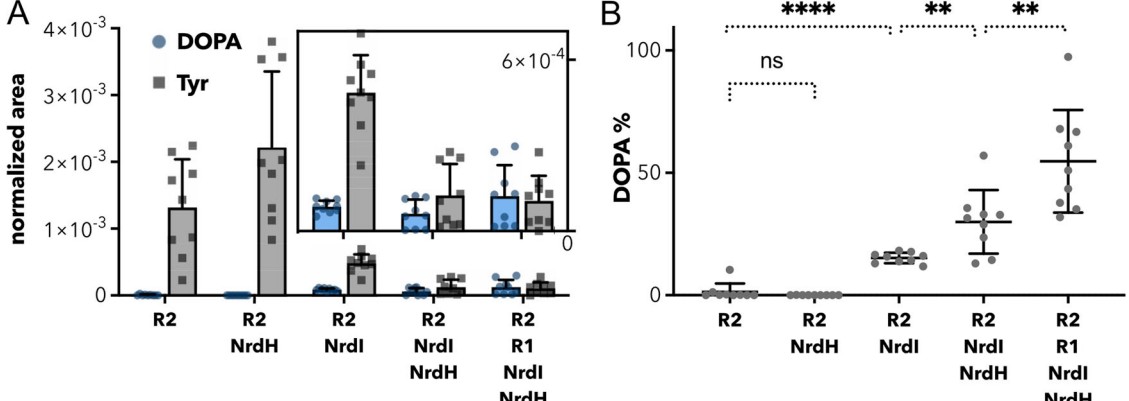

**Fig. 3 | Mass spectrometry of *Bv*R2e$_{QSK}$ produced in five different conditions of coexpression. A** Integrated MS1 peak area of peptides containing residue 150 normalized to protein amount. Individual data points are shown as grey squares for tyrosine and blue dots for DOPA containing fragments. The mean of nine measurements is represented as a bar, the standard deviation is given by a line above the bar. The inset shows a magnification of the bars below for better visualization of individual data points. **B** Percentage of DOPA-containing peptides of all measured peptides containing residue 150. Mean and standard deviation are shown as lines. A significant increase of DOPA formation between different expressions is marked with stars (exact values can be found in the main text). Raw data for both panels can be found in Supplementary Data 1.

corresponding R1 from *B. vaginale* was heterologously expressed and purified. An in vitro assay was set up containing the proteins *Bv*R1 and *Bv*R2, the effector ATP and the substrate CDP and incubated for 30 min at room temperature. As a positive control active R2e$_{VPK}$ from *Mesoplasma florum* was used in combination with its corresponding R1. Once active, *Mf*R2e houses a stable radical that can be used for several turnovers and thus does not require NrdI[14]. The formation of the product dCDP was probed by applying the mixture to an HPLC column. R2e$_{QSK}$ with unmodified tyrosine and DOPA containing R2e$_{QSK}$ were tested (purifications a) and e) from Fig. 2). The R1-R2$_{QSK}$ complex was not able to reduce CDP, regardless of whether the R2e$_{QSK}$ used contained tyrosine or DOPA. This confirms the observation that the R2e$_{QSK}$ seems to be inactive after purification despite DOPA formation. NrdI is suggested to deliver the oxidant for radical formation in the active site. Therefore, chemically reduced NrdI was added prior to the reaction start. After reoxidation of NrdI in air dCNPs were formed by the RNR complex. The same reaction with unmodified R2e$_{QSK}$ did not yield any CDP reduction (Fig. 4). The in vitro activity of RNR$_{QSK}$ is therefore dependent on the presence of DOPA in R2 and reduced NrdI in an aerobic environment. These results demonstrate that the QSK substituted R2 metal site provides a functional protein for ribonucleotide reduction.

**Structures of *Bv*R2**

To examine the structural details of unmodified and modified, activatable *Bv*R2e$_{QSK}$ we crystallized the purified protein and solved two crystal structures. We first crystallized R2 expressed alone and solved the structure of unmodified *Bv*R2e$_{QSK}$ (*Bv*R2$_{QSK}$-tyr) at 1.8 Å resolution in space group $P3_121$ with four monomers in the asymmetric unit (Table 1) forming two

homodimers that correspond to the physiological active form of other R2 subunits (Fig. 5A). Of the 364 amino acids, residues 6 to 333 could be modelled in the electron density.

The protein forms the characteristic ferritin-fold with a bundle of 4 alpha-helices that houses the active site[1]. The C-terminus is unstructured as is typical for R2 proteins. The N-terminus is extended and contacts the second monomer of the dimer, creating a large interaction surface of about 4300 Å$^2$ and forming 6 salt bridges between the monomers (Fig. 5B). The typical interaction surface for published structures of R2 is between 1000 and 2500 Å$^2$; the area for *Bv*R2$_{QSK}$ is therefore much larger (Supplementary Table 3). The extension of the N-terminus is on the opposite side of the protein where R1 and NrdI would putatively bind and is hence not expected to influence those interactions (Supplementary Fig. 1).

The active site residues are well-defined. There are no metal ions observed in the active site. Residues Y150, H146, H240 and D112 are in the same position as in other R2bs. The remaining glutamates that usually bind metals in the site are as expected from the sequence substituted to Q143, S203 and K237. The position of Mn1 is occupied by the ε-ammonium group of K237 and in the position of Mn2 a well-defined water is bound (Fig. 5C).

The second crystal structure we solved was from R2 protein coexpressed with R1, NrdI and NrdH, hereafter called *Bv*R2e$_{QSK}$-DOPA. Crystals belong to the same space group with similar unit cell dimensions and resolution (Table 1). The two structures are similar overall with an Cα root-mean-square deviation (RMSD) value of 0.16 Å. The conserved tyrosine close to the active site shows an additional density, corresponding to a modification to a DOPA. Composite omit maps confirmed that the tyrosine is modified between structures (Fig. 5D). Overlaying both structures shows

**Fig. 4 | Activity assay results.** Each reaction contained R1 and R2. $Mf$RNR$_{VPK}$ was used as a positive control. $Bv$R2$_{QSK}$ was tested with either an unmodified tyrosine ($Bv$R2-tyr) or DOPA ($Bv$R2-DOPA) in the active site. The RNRs were tested with and without freshly reoxidized $Bv$NrdI present. CDP was used as a substrate, dCDP was formed during catalysis. DTT was added as a reductant to prevent R1 from oxidizing. Raw data for the graph can be found in Supplementary Data 1.

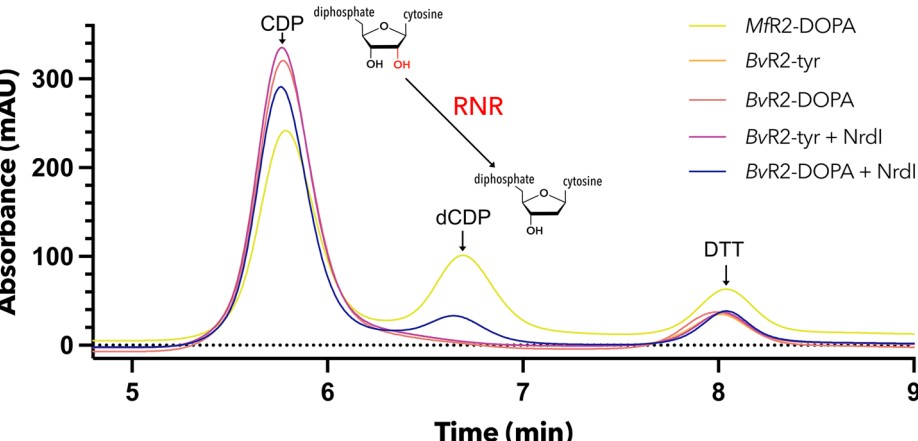

no further structural reorganization; all other active site residues are in the same position (Fig. 5D, Supplementary Fig. 2).

We compared the DOPA-modified R2e$_{QSK}$ structure with structures of DOPA containing R2e$_{VPK}$. For this protein, structures of two different activation states are available: a structure of the active, radical harbouring R2, $Mf$R2e$_{VPK}$-active (PDB ID: 8BT3) and a structure of the radical-lost ground state, $Mf$R2e$_{VPK}$-ground (PDB ID: 8BT4)[15]. These three structures superimpose generally well. The conformation of the active site residues DOPA150, D112 and K237 aligns best with the radical-lost ground state of $Mf$R2e and differs from the radical state, suggesting that we observe an inactive state of the protein in our structures. However, the hydrogen bonding network and water structure in the active site between the VPK and QSK versions of R2 are clearly different due to the bulkier Q and S compared to V and P (Supplementary Fig. 3). This will likely contribute to the differences in activity between R2e$_{VPK}$ and R2e$_{QSK}$.

## Discussion

This work investigates R2e$_{QSK}$, a class of RNR that evolved from R2b and diverged further into R2e$_{VPK}$. In both R2e variants, the typically metal-binding glutamates of other R2 subclasses are substituted with residues not known to act as metal ligands[14]. In R2 subclasses a-d, a dinuclear iron and/or manganese centre is mandatory for radical formation. Previously it has been shown for R2e$_{VPK}$ that it indeed does not bind a metal cofactor but still can generate a radical[13,14]. R2e therefore must employ a different mechanism of radical formation than the metal-dependent R2 subclasses. In R2e$_{VPK}$ the radical harbouring residue is not a tyrosine like in R2b but a post-translationally generated DOPA. The post-translational modification is facilitated by the coexpression of NrdI. Furthermore, the simultaneous expression of NrdI and R2e$_{VPK}$ leads to an active, radical-containing protein that is stable for hours in vitro[14]. Here, we investigated R2e$_{QSK}$, the evolutionary link between the metal-dependent R2b and the metal-free R2e class. We could demonstrate that R2e$_{QSK}$ likewise does not contain any iron or manganese. We found the homologous tyrosine modified similarly to a DOPA in R2e$_{QSK}$. Like in R2e$_{VPK}$, DOPA is only formed with the coexpression of NrdI. To our surprise, we found an increase of DOPA formation with simultaneous expression of R2, NrdI and NrdH and the highest level of modification was observed with expression of the whole operon. In other R2 classes, neither R1 nor NrdH have been considered to be involved in radical formation or cofactor maturation. We investigated the gene neighbourhood in organisms encoding R2e$_{VPK}$ or R2e$_{QSK}$. $nrdH$ has not been found near $nrdF_{VPK}$ in any genomes and is found near $nrdF_{QSK}$ only in certain genomes, although it is a short gene that can be overlooked. NrdH could therefore act as an enhancer for DOPA formation but does not seem to be mandatory. UV-Vis as well as structural data suggests that the protein is in the non-radical state after purification in contrast to R2e$_{VPK}$. It is unknown if the $meta$-hydroxylation of the tyrosine is mechanistically connected with radical generation or if they are two separate processes. However, we could show in vitro activity of class Ie$_{QSK}$ and therefore that it is a functional RNR

system, which must be able to generate a radical. Catalysis could only be observed with the DOPA-containing protein. As in R2b, the activity required the presence of reduced NrdI. Significantly less product was formed with R2e$_{QSK}$ than with the control, R2e$_{VPK}$. For R2e$_{VPK}$ it is known that, once activated, the protein can be used for multiple turnovers[14]. The presence of NrdI to provide an oxidant for radical generation is therefore not required in an in-vitro activity assay. A short-lived radical on the other hand would require a constantly available pool of reduced NrdI to provide the oxidant for radical generation. Under the conditions we used, this pool would be very limited and could explain the lower activity. Since we currently do not have a protocol that allows constant replenishment of reduced NrdI, we cannot exclude it as the limiting factor in the activity assays and therefore did not try to quantify the RNR$_{QSK}$ activity or compare it to other RNRs. The structural comparison of both proteins shows additional waters and an extended hydrogen bond network in the VPK ground state protein compared to the QSK site. This might impact the path of the oxidant required for the radical formation or the DOPA environment and the stability of the radical after generation. These results open interesting avenues for further study.

Here we show that proteins of the class Ie QSK group evolutionarily bridging the metal-containing class Ib and class Ie VPK and serving as the only aerobic RNRs in several organisms, do indeed form functional RNR systems.

## Materials and Methods
### Bioinformatics
RNR proteins were identified in all species representative genomes from GTDB release 08-RS214[25] using a set of inhouse HMMER[26] version 3.3 profiles for RNR R2 proteins. Prior to identification of RNR proteins, the genomes were annotated with Prokka[27] version 1.14.6. Subsequently, R2e$_{QSK}$ and R2e$_{VPK}$ sequences were identified in alignments by searching for the QSK and VPK substitutions respectively. The species encoding genes for R2e were plotted on the GTDB phylogeny using Anvi'o version 8.1[28].

Gene neighbourhoods were identified in the same set of GTDB species representative genomes by identifying RNR genes separated by no more than 300 nucleotides on the same DNA strand.

### Cloning
Several genetic constructs were used in this work; all use the genomic DNA of *Bifidobacterium vaginale* ATCC 14019 as a template:

pET28_TEV_B$v$nrdF: The gene encoding *Bifidobacterium vaginale nrdF* was PCR-amplified from genomic DNA (DSMZ, DSM 4944) using the $Bv$R2 forward primer (5′-GTA*CATATG*ACAGAACTCAGCCCAACT GCG-3′) and $Bv$R2 reverse primer (5′-GTA*GGATCCC*TAGAAATCCC AATCATCGTCATC-3′). The resulting 1.1 kb PCR product was digested with NdeI and BamHI (Thermo Fisher Scientific), and then ligated into a modified pET-28a plasmid (Novagen), with a tobacco etch virus (TEV)

## Table 1 | Data collection and refinement statistics

| PDB ID | $Bv$R2e$_{QSK}$-tyr 8RAG | $Bv$R2e$_{QSK}$-DOPA 8RAH |
|---|---|---|
| **Data collection statistics** | | |
| Synchrotron/Beamline | Diamond/i04 | MaxIV/Biomax |
| Wavelength (Å) | 0.95374 | 0.9763 |
| Space group | P3$_1$21 | P3$_1$21 |
| Unit cell dimensions a, b, c (Å) | 173.5, 173.5, 157.6 | 173.7, 173.7, 157.9 |
| Unit cell angles α, β, γ (°) | 90, 90, 120 | 90, 90, 120 |
| Resolution range (Å) | 47.74 - 1.9 (1.968 - 1.9) | 49.68 - 1.9 (1.968 - 1.9) |
| Unique reflections | 214258 (21312) | 215066 (21342) |
| Multiplicity | 21.0 (21.5) | 16.9 (13.7) |
| Completeness (%) | 99.95 (99.83) | 99.63 (96.60) |
| Mean I/sigma (I) | 12.94 (0.68) | 13.20 (0.40) |
| Wilson B-factor (Å²) | 36.95 | 45.56 |
| $R_{merge}$ (%) | 17.04 (223.9) | 13.16 (458.4) |
| $R_{meas}$ (%) | 17.46 (229.3) | 13.57 (476.1) |
| $R_{pim}$ (%) | 3.80 (49.27) | 3.28 (127.5) |
| CC$_{1/2}$ | 0.999 (0.596) | 0.999 (0.271) |
| C* | 1 (0.864) | 1 (0.653) |
| **Refinement statistics** | | |
| Resolution range used in refinement (Å) | 47.74 - 1.9 (1.97 - 1.9) | 49.68 - 1.9 (1.97 - 1.9) |
| Reflections used in refinement | 214176 (21277) | 214297 (20616) |
| Reflections used for $R_{free}$ | 1450 (141) | 1839 (177) |
| $R_{work}$ (%) | 16.66 (30.45) | 17.11 (39.42) |
| $R_{free}$ (%) | 19.44 (31.77) | 18.92 (42.60) |
| RMSD, bond distances (Å) | 0.006 | 0.007 |
| RMSD, bond angles (°) | 0.74 | 0.76 |
| Ramachandran favored (%) | 98.62 | 98.44 |
| Ramachandran allowed (%) | 1.08 | 1.25 |
| Ramachandran outliers (%) | 0.31 | 0.31 |
| Rotamer outliers (%) | 1.26 | 1.71 |
| Clashscore | 1.82 | 1.70 |
| Protein residues | 6 - 333 (of 364) | 8 - 332 (of 364) |
| Average B-factor (Å²) | 42.47 | 51.91 |
| macromolecules | 41.69 | 51.78 |
| solvent | 49.55 | 53.74 |
| Number of non-H atoms | 12038 | 11448 |
| macromolecules | 10842 | 10694 |
| solvent | 1196 | 754 |

Data in parenthesis are for the highest resolution shell.

protease site instead of the thrombin cleavage site, immediately following a N-terminus 6xHIS-tag.

pET28_Bv*nrdHIEF*: The plasmid was similarly generated using the *Bv*O forward primer (5′-GTA*CCATGG*TGACTATTACAGTTTTCACCA AGCC-3′) and *Bv*O reverse primer (5′-GTATTA*CTCGAG*GAAATCCC AATCATCGTCATCGG-3′) to amplify a 4.3 kb product from the genome. The PCR product was digested with NcoI and XhoI (Thermo Fisher Scientific), and then ligated into the modified pET-28a plasmid, placing a C-terminal uncleavable 6xHIS-tag on the *Bv*R2 protein.

pET28_TEV_Bv*nrdFI*: The plasmid was generated by ligating the PCR product of *Bv*NrdI forward primer (5′-GTA*GGATCC*GTAAAGGAGAT ATACCATGTGTGATCAAGAAAATATGTCAC-3′) and *Bv*NrdI reverse

primer (5′-ACT*GAGCTC*TTATTTCGTTTCTTCTGTTTCTTTTAATT GC-3′) from the genome into the pET-28a_Bv*nrdF* plasmid between the BamHI and SalI (Thermo Fisher Scientific) restriction sites.

pGEX-4T-1-M_Bv*nrdH*: The plasmid was ordered from Genscript. The gene is inside of an NdeI and BamHI restriction site and contains a TEV protease cleavage site on the N-terminus; the plasmid has an N-terminal GST-tag.

pET28_TEV_Bv*nrdE*: The plasmid was generated by ligating the PCR product of *Bv*NrdE forward primer (5′-GTA*CATATG*AATAGCACGG ATCCGATGAGC-3′) and *Bv*NrdE reverse primer (5′-GTA*CTCGAG*TT ACAGCGTGCAGCTTACGCAAC-3′) from the genome into the pET-28a plasmid between the Nde1 and Xho1 (Thermo Fisher Scientific) restriction sites.

### Protein Expression and Purification
**Expression and purification of *Bv*R2s.** Five different protein expressions were used for this work. *Escherichia coli* BL21(DE3) cells were transformed with the following scheme (see also Fig. 2a):

1. R2e alone: pET28_TEV_Bv*nrdF*
2. R2e + NrdH: pET28_TEV_Bv*nrdF* + pGEX-4T-1-M_Bv*nrdH*
3. R2e + NrdI: pET28_TEV_Bv*nrdFI*
4. R2e + NrdI + NrdH: pET28_TEV_Bv*nrdFI* + pGEX-4T-1-M_Bv*nrdH*
5. R2e + NrdI + NrdH + R1: pET28_Bv*nrdHIEF*

The pET28 constructs were selected with kanamycin and the pGEX-4T-1-M with ampicillin/carbenicillin. The appropriate combination of antibiotics was used to ensure successful selection of cells containing the desired plasmids. For each expression, an LB-media (Lysogeny Broth medium) preculture was inoculated with transformant and grown overnight at 37 °C. The following day, large-scale cultures were prepared, 0.5% (vol/vol) of the preculture in 1.6 ml of TB (Terrific Broth medium) supplemented with the correct antibiotics (100 µg/ml carbenicillin and/or 50 µg/ml kanamycin) and 1:10,000 (vol/vol) antifoam 204. The cultures were grown at 37 °C in a benchtop bioreactor system (Harbinger) to an OD$_{600}$ of about 1, subsequently cooled to room temperature and induced with 0.2 M isopropyl β-D-1- thiogalactopyranoside (IPTG). The cells were harvested the next day by centrifugation and flash-frozen in liquid nitrogen.

All proteins were purified following a similar protocol as described in Srinivas et al., 2018[14]. In short, about 20 g of cells were lysed with a sonicator, Sonics VCX130 (Sonics) after thawing them and adding DNAse and a tablet of protease inhibitor in lysis buffer (50 mM Tris pH 7.5, 150 mM NaCl). The crude extract was cleared by centrifugation and NiNTA agarose resin in a gravity flow column equilibrated with lysis buffer. The clear lysate was applied to the column, washed with lysis buffer and eluted with elution buffer (50 mM Tris pH 7.5, 150 mM NaCl, 250 mM imidazole). The eluate was concentrated to 5 ml and loaded on a HiLoad 16/60 Superdex 200 size-exclusion column (SEC) (GE Healthcare) attached to an ÄktaPrime Plus (GE Healthcare) equilibrated with SEC buffer (50 mM Tris pH 7.5, 25 mM NaCl). The SEC elution was analysed via SDS gel electrophoresis and pure fractions of the correct size were pooled. To constructs containing a TEV cleavage site, his-tagged TEV protease was added in a molar ratio of 1:50 and incubated overnight at 4 °C. Uncleaved protein and TEV protease were removed by another NiNTA step with the pure protein in the flow-through. The protein was concentrated to about 20 mg/ml, aliquoted and flash-frozen in liquid nitrogen. The protein was stored at −80 °C until further use.

**Expression and Purification of *Bv*R1.** Expression of *Bv*R1 was conducted as described above for R2, with the change of expression temperature and time of the large-scale culture being 37 °C for 3 hours. The purification was also conducted in the same way as described above.

**Expression and Purification *Mesoplasma florum* R1 and R2.** Both proteins were used in the activity assay as a positive control. The proteins were expressed and purified as described in Srinivas et al.[14].

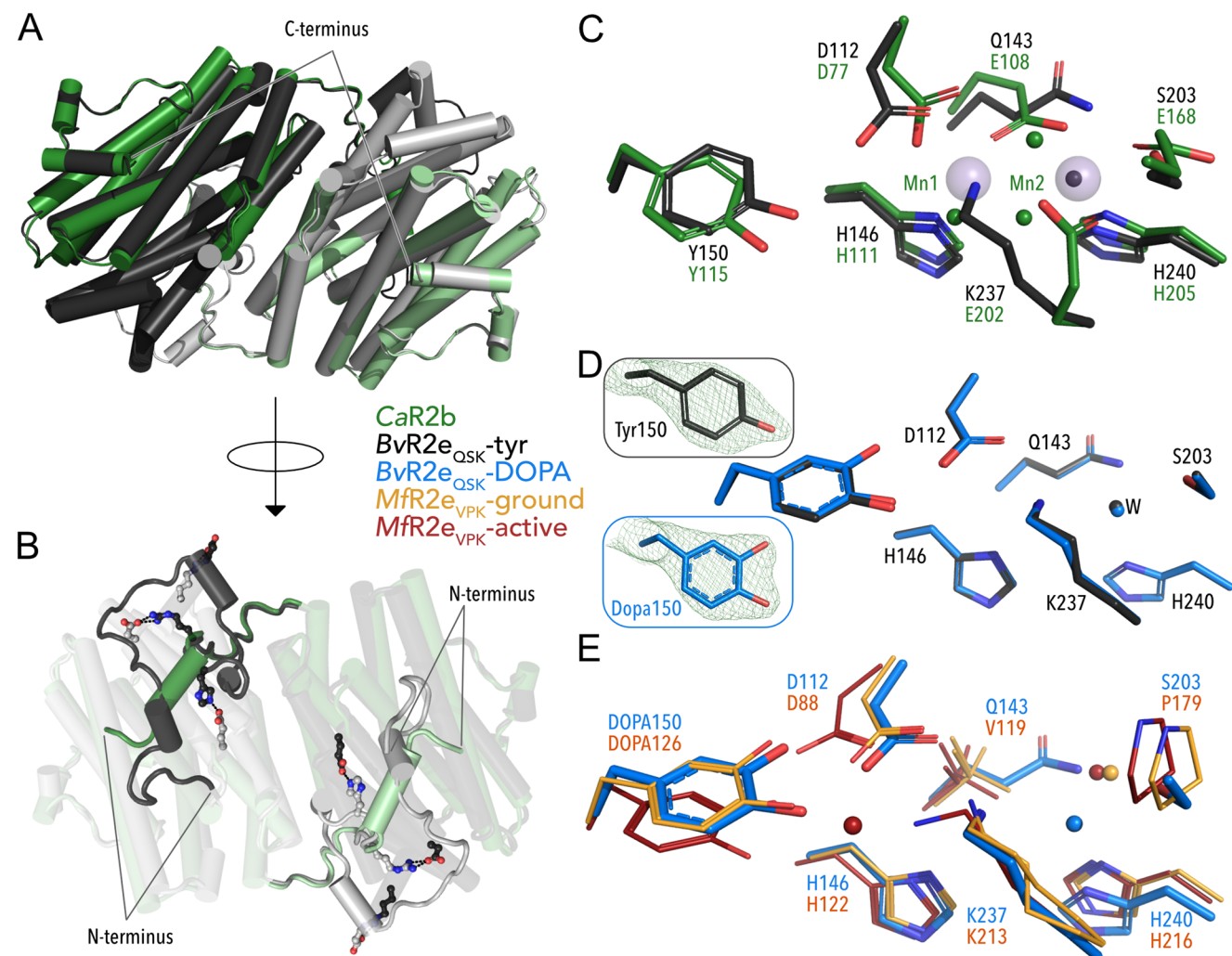

**Fig. 5 | Structure comparison of *Bv*R2e_QSK.** **A** Overall structure of the homodimer with each chain in light and dark grey, compared to *Corynebacterium ammoniagenes* R2b in green (*Ca*R2b, PDB ID: 3MJO). **B** 180° turn of the view in **A**. The extended N-terminus of *Bv*R2e is compared to *Ca*R2. Residues involved in salt bridges in *Bv*R2 are shown as stick and ball. **C** Active site residues of *Bv*R2e in grey overlayed with *Ca*R2b residues in green. The two manganese ions bound in R2b are shown as transparent spheres. Waters are shown as small spheres coloured the same as the corresponding structure. **D** Comparison of the active site of *Bv*R2 expressed alone in grey with the active site of *Bv*R2e coexpressed with the whole operon in blue. Composite omit maps of residue 150 are shown for each structure contoured at 2 σ. **E** Comparison of *Bv*R2e_QSK-DOPA with *Mf*R2e_VPK containing the DOPA modification in the radical (red) and ground (yellow) state. (PDB IDs: 8BT3 = radical state, 8BT4 = ground state).

## Total reflection X-ray fluorescence

The five different purifications of R2 were analysed for their metal content. Each purification was measured in triplicate. The proteins were concentrated to about 150 – 600 µM in SEC buffer. As an internal standard, 20 mg/l gallium in distilled water was used; 7 µl of standard was mixed with 7 µl sample. Of the mixture, 10 µl were transferred to a clean siliconized quartz sample carrier and dried on a hot plate. The samples were measured with a Bruker PicoFox S2 spectrometer (Bruker) for 1000 s each. The results were analysed with the Bruker Spectra software version 7.8.2.0 that was provided with the instrument.

## Liquid chromatography mass spectrometry

Five different protein preparations of similar concentration were analysed by LC-MS. Each sample was prepared in triplicate, so that a total of 15 samples were prepared, and each triplicate was measured three times in LCMS to overcome variations in LCMS sensitivity over time. Approximately 50 µg of protein (or 3 µL of the original protein solution, which had a protein concentration of approximately 400 µM) was used in each individual sample preparation, and samples were diluted in 100 µL of lysis buffer (50 mM Ammonium Bicarbonate buffer pH 7.6, 1 mM Dithiothreitol (DTT)). After heating at 95°C in a heater block with mild

shaking for 15 min, samples were allowed to cool down, and 10 µL of 0.04 M chloroacetamide was added (approx. final conc. 4 mM) to alkylate the cysteine residues. Afterwards, potential contaminants were removed using SP3 magnetic beads following a modified version of the SP3 protocol[29,30].

Briefly, the bottles of Sera-Mag Speed beads (Cytiva P/N 65152105050250 and P/N 45152105050250) were shaken gently until no deposit was observed, and 50 µL was taken from each bead type into one tube. The 100 µL of beads were washed 3x with water (Optima LCMS grade P/N W6-212 from Fisher Scientific) by repeating the following steps 3x: placing the tubes on the magnetic rack, allowing beads to attach for 30 s, removing and discarding all liquid from the beads, taking the tube out from the magnetic rack, and adding 500 µL H₂O with gentle pipetting mixing. The final 500 µL beads solution was the stock solution and was always gently mixed by pipetting prior to drawing a 25 µL aliquot to add to each sample. After adding the SP3 magnetic beads to the samples with gentle pipetting mixing, 250 µL of acetonitrile was added, and then the tubes were placed in the rotating incubator with gentle mixing for 20 min. Thereafter, tubes were placed in the magnetic rack, and beads were allowed to settle for 2 min. All liquid was removed and discarded. There followed 2 washes with 200 µL of 70% EtOH and a final wash with 200 µL of acetonitrile. The dry beads with

bound proteins were reconstituted in 90 μL of trypsin solution (50 mM Ammonium Bicarbonate buffer pH 7.6, 2 μg trypsin) and gently mixed by pipetting. Then 10 μL of acetonitrile was added and incubation at 37 °C with mild shaking followed for 14 h. The samples were then placed in the magnetic rack, and after 2 min the liquid from each sample was collected and transferred to a fresh tube. The samples were again placed in the magnetic rack, and after 2 min the liquid from each sample was collected and transferred to an LCMS vial (Waters P/N 186000384 C). The vials were then dried in a speed vac. To each vial, 100 μL of 10% formic acid solution in LCMS grade water (Optima LCMS grade P/N W6-212 from Fisher Scientific) was added to dissolve the peptides.

From each sample LC-MS run triplicates (yielding 45 LCMS raw files) were done, the autosampler (Ultimate 3000 RSLC system, Thermo Scientific Dionex) injecting 1 μl into a C18 trap desalting column (Acclaim pepmap, C18, 3 μm bead size, 100 Å, 75 μm × 20 mm, nanoViper, Thermo). After 6 min of flow at 5 μL/min with the loading pump, the 10-port valve switched to analysis mode in which the NC pump provided a flow of 250 nL/min through the trap column. The curved gradient (curve 6 in the Chromeleon software) then proceeded from 3% mobile phase B (90% acetonitrile, 5% DMSO, 5% water, 0.1% formic acid) to 45% B in 50 min followed by a wash at 99% B and re-equilibration. Total LC-MS run time is 24 min longer than the gradient time. We used a nano EASY-Spray column (pepmap RSLC, C18, 2 μm bead size, 100 Å, 75 μm × 50 cm, Thermo) on the nanoelectrospray ionization (NSI) EASY-Spray source (Thermo) at 60°C. Online LC-MS/MS was performed in DDA (data dependent acquisition) mode using a hybrid Q-Exactive HF mass spectrometer (Thermo Scientific). FTMS master scans with 120,000 resolution (and mass range 300-1500 m/z) were followed by data-dependent MS/MS (60,000 resolution) on the top 5 ions using higher energy collision dissociation (HCD) at 30% normalized collision energy. Precursors were isolated with a 2 m/z window and an isolation offset of 0.5 m/z. Automatic gain control (AGC) targets were 1e6 for MS1 and 1e5 for MS2. Maximum injection times were 100 ms for MS1 and MS2. The entire duty cycle lasted ~1 s. Dynamic exclusion was used with 30 s duration. Precursors with charge states 2-7 were included. An underfill ratio of 1% was used. All chemical reagents are LCMS or Bio ultra-grade and purchased from Sigma Aldrich/Merck unless stated otherwise.

The 45 LCMS raw files were processed in PEAKS Studio 11 (build 20230414, BSI - Bioinformatics Solutions Inc), each file being quantified individually. The enzyme was trypsin, the instrument was Orbitrap (Orbi-Orbi), fragmentation method HCD and acquisition DDA. The processing workflow was a PEAKS Q (de novo assisted Quantification), which began with a Data Refine Step with Chimera feature association, then a Denovo step with a precursor mass error tolerance of 10 ppm, a fragment mass error tolerance of 0.02 Da, carbamidomethylation (C) as fixed modification, and deamidation (NQ) and Oxidation (M) as variable modifications, with a maximum of 3 variable modifications per peptide. There followed a DB search against a concatenated database of 5559 protein entries including sequences from *E. coli* K12 and *Bifidobacterium vaginale* proteomes from uniprot.org plus the ribonucleotide reductase QSK R2e variant and the NrdI isoform sequences produced recombinantly for these experiments. A maximum of 3 missed cleavages was allowed, digest mode was specific, peptide length was 6-45 aas, and Deep Learning Boost was used. The other settings were the same as in the Denovo step. The report filters were PSM FDR 1%, -10LgP >= 15, Protein Unique Peptides >= 1, Denovo Only ALC(%) >= 50, Denovo Only Fully Matched = No. For the final step, the Label Free Quantification, settings employed were: Mass Error Tolerance of 20.0 ppm, Retention Time Shift Tolerance was Auto detect, Feature Intensity > 100000, 0 < RT < max, Base Sample was average. Peptide Feature Filters were: Avg. Area >= 200000, Quality >= 20.0, Peptide Id Count >= 0 per group, 2 <= Charge <= 5. Finally, TIC was used as normalization method.

Measurements for each protein purification were compared against other purification with an unpaired t-test and significant differences were calculated in GraphPad Prism (Version 10.2.2 for MacOS).

All MS raw data and search results files have been deposited to the ProteomeXchange Consortium via the PRIDE partner repository with the dataset identifier PXD045227.

## Activity assay

For the activity assay R1 was mixed in a buffer cocktail containing HEPES pH 8.0, 20 mM magnesium chloride and 25 mM dithiothreitol (DTT). ATP (1 mM) and CDP (2 mM) were used as effector and substrate combination. 50 μl reactions were carried out for all the R1-R2 combinations. R2 was added in the end to start the reaction. Reaction was allowed to carry on for 30 minutes at room temperature, after that 50 μl of 100% methanol was added to stop the reaction. 100 μl of distilled water was added to the mixture. Samples were centrifuged at 20000 g for 5 minutes to remove all the proteins. Finally, 10 μl of the mixture was loaded on HPLC (Agilent) using an Agilent ZORBAX RR StableBond (C18, 4.6 × 150 mm, 3.5 μm pore size) equilibrated with buffer A (10% methanol, 50 mM potassium phosphate buffer, pH 7, 10 mM tetrabutylammonium hydroxide). A sample of 10 μL was injected and eluted at 1 mL/min with a gradient of buffer B (30% methanol, 50 mM potassium phosphate buffer, pH 7, 10 mM tetrabutylammonium hydroxide). For all assays, 2 μM R1 was used, while R2 concentration was varied between 2 and 4 μM. To generate a radical, *Bv*R2e (100 μM) was first mixed with an equimolar amount of *Bv*NrdI. 2 mM sodium dithionite was then added to reduce NrdI. After 10 minutes of incubation, the mixture was re-oxidized by pipetting.

## Crystallization

Crystallization conditions for *Bv*R2 have been screened with the Morpheus crystallization screen (Molecular Dimensions). Many conditions gave rise to crystals but the best diffracting crystals were produced by condition G4 (0.1 M Carboxylic acids, 0.1 M Buffer System 1 pH 6.5 37.5% v/v Precipitant Mix 4; Carboxylic acids = 0.2 M Sodium formate; 0.2 M Ammonium acetate; 0.2 M Sodium citrate tribasic dihydrate; 0.2 M Potassium sodium tartrate tetrahydrate; 0.2 M Sodium oxalate; Buffer System 1 = 1 M Imidazole, 1 M MES monohydrate (acid), pH 6.5; Precipitant Mix 4 = 25% v/v MPD; 25% PEG 1000; 25% w/v PEG 3350). The protein was crystallized at 25 mg/ml by sitting-drop vapour diffusion in an MCR 2-well crystallization place (SwissSci) with a 1:1 ratio of protein to crystallization condition. For *Bv*R2e_QSK-tyr, the protein used was expressed alone and was crystallized without further optimization of the crystallization condition. The crystal with the best resolution was at the time of data collection two month old while newer crystals diffracted poorer. For *Bv*R2e_QSK-DOPA, the protein used was coexpressed with the whole operon, with an uncleavable C-terminal his-tag, and further optimization was required for crystallization. The Additive Screen (Hampton) gave the best results with an addition of 4% tert-butanol. The crystal used for structure solution was at the time of data collection four months old (fresh crystals resulted in lower resolution). Crystals were flash-frozen in liquid nitrogen without the addition of any cryoprotectant.

## Data collection and structure determination

The crystal from the modified protein was collected at the Biomax beamline at MaxIV (Lund, Sweden) and the unmodified protein crystal at the i04 beamline at Diamond Light Source (Oxfordshire, UK). Data was reduced with XDS[31]. *Bv*R2 crystal structures were solved by molecular replacement using PHASER[32] with a previously solved, unpublished *Bv*R2 as a starting model. The suggested solutions for both proteins were in space group P3$_1$21 with 4 monomers in the asymmetric unit (Table 1). The datasets were refined with phenix.refine[33], built in coot[34] and validated with MolProbity[35]. Refinement of all atoms included isotropic B-factors, TLS parameters, occupancy and reciprocal space refinement. The final high-resolution cut-offs for the data sets were decided by using PAIREF[36]. A starting resolution cut-off of 2.1 Å was chosen and paired refinement for 0.1 Å increment was performed. PAIREF suggested the inclusion of both the 2.1-2.0 Å and the 2.0 - 1.9 Å shell for the final model, mostly based on the improvement of Rfree with the inclusion of those shells. Waters were initially added using phenix.refine

and in later refinements manually adapted. A composite omit map with simulated annealing for each model was calculated with the phenix suite version 1.20.1[37]. The refined structures were compared and RMSD values calculated with SSM superimpose[38]. Molecular figures were created with PyMOL Molecular Graphics System, Version 2.4.2 Schrödinger, LLC. The molecular surface areas and salt bridges listed in supplementary Table 3 were determined by PDBsum[39]. The structural formula in Fig. 4 was created with ChemDraw and all figures were assembled in Affinity Designer.

### Statistics and Reproducibility

For TXRF metal quantification, the five different R2 protein purifications were measured in triplicates and the mean and standard deviation for each purification were calculated. All the measured values, means and standard deviations are presented in Supplementary Table 2.

For LC-MS, three replicates were prepared for each of the five different R2 protein purifications, and each sample was measured in triplicates. The mean and standard deviation were calculated for each sample. Then measurements for each protein purification were compared against other purifications with an unpaired t-test and significant differences were calculated in GraphPad Prism. The means and standard deviations are shown in Fig. 3 A, the results of the t-test in Fig. 3 B. The data the figures are based on can be found in Supplementary Data 1.

### Reporting summary

Further information on research design is available in the Nature Portfolio Reporting Summary linked to this article.

### Data availability

Two crystal structures have been deposited in the Protein Data Bank with the following IDs: 8RAG for the tyrosine containing R2e$_{QSK}$ form of *Bifidobacterium vaginale*, and 8RAH for the active, DOPA-containing form. All MS raw data and search results files have been deposited to the ProteomeXchange Consortium via the PRIDE partner repository with the dataset identifier PXD045227.

Other data used in the manuscript can be requested from the corresponding authors.

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

## Acknowledgements

We would like to thank Inna Rozman Grinberg for help with the activity assays. We would like to thank the staff at the MaxIV synchrotron (Lund, Sweden), especially the staff of the Biomax beamline and Diamond Light Source, with special thanks to the staff at the i04 beamline for support with the data collection. Financial support was provided by the Swedish Research Council (2021-03992) and the Knut and Alice Wallenberg Foundation (2017.0275 and 2019.0436). DL acknowledges financial support from Britt-Marie Sjöberg (Swedish Research Council, grant number 2019-01400) and all authors acknowledge her for input on the research.

## Author contributions
J.J. performed protein expression, TXRF and UV-Vis experiments and analysis, performed LC-MS, crystallization experiments and analysis, manuscript - initial draft, created all figures except Fig. 1 D.L.: performed organismal distribution and genetic analyses, creation of Fig. 1 R.M.B.: supervised LC-MS experiment and analysed the data RK: performed and analysed activity assays V.S.: conceptualization, cloned and tested initial constructs H.L.: conceptualization, supervision, study development M.H.: conceptualization, supervision, study development. All authors contributed to the review and editing of the initial draft.

## Funding

## Competing interests
The authors declare no competing interests.
