## [Transparent Peer Review file · Communications Biology]

Characterization of a second class le ribonucleotide reductase

Corresponding Author: Dr Hugo Lebrette

This manuscript has been previously reviewed at another journal. This document only contains information relating to versions considered at Communications Biology.

Version 0:

Reviewer comments:

Reviewer #2

(Remarks to the Author)

Noteworthy results

In our previous review of this work, we noted that while most of the experiments performed were sound, we hoped for a more in-depth characterization of the class le QSK RNR to increase the overall noteworthiness of the manuscript. The authors have addressed our questions of interpretation, fixed minor issues or errors, and explained in their responses to the first review why the further experiments we suggested might be challenging. Because they did not perform further experiments, the significance of the results has not increased substantially. As this manuscript is now submitted to Communications Biology, we have tried to compare to other papers published in this journal. We feel that the limited quantitative results of the enzymology in this work is a little below the norm for this journal, but may be acceptable due to the challenges of developing a quantitative assay for the enzyme of interest.

Significance of work

This work is of good merit, and as noted in our previous review, it is of significance to the field of enzymology and RNR diversity. The authors have updated text in the manuscript to more clearly define what was previously known in Srinivas et al. 2018 and how this paper is building upon this. The introduction is still a bit confusing to people unfamiliar with RNRs. One potential suggestion: a figure in the introduction that visually communicates the mechanism and purpose of each subunit would benefit the reader, as the written description in lines 39 to 56 is verbose.

Does the work support the conclusions and claims?

The authors have addressed most of our questions for the following two points:

1. We asked for further activity characterization and more compelling evidence of the DOPA radical. The authors addressed our questions in their "response to reviewers" and explained why quantitative activity assays and EPR or UV/vis experiments would be challenging under current circumstances. This is understandable, but it does make for a less compelling manuscript.

Fundamentally, we are curious about the cause of what seems to be overall low activity of the protein of interest as compared to prior RNRs, including that characterized by Srinivas et al. We would like to confirm that Figure 2c iron content is in fact reported in %, as written (e.g., 0.33%), and not in mol/mol fraction (e.g., 0.33 or 33%). Were it the latter, we would wonder if the RNR is in fact a metalloprotein with poor holo-protein content.

Minor note: We feel that inclusion of the activity assay results (Supplemental Figure 1) in the main text would be relevant to show the reader that the purified proteins are active for ribonucleotide reduction. The figure could be paired with a scheme of the chemical reaction, which would help the reader follow the section "In vitro activity of BvRNRQSK" (lines 225 to 244). The noteworthy conclusion that the DOPA modification and NrdI are required for RNR activity is substantiated by the data, and therefore it would benefit the reader to see the data in the main text.

2. We posed questions about the data quality of the 1.9 Å crystal structure with a DOPA radical. The authors have responded with a reasonable explanation for their resolution cut-off that was used. We do still wonder about the occupancy of the DOPA. The text describes it as clearly DOPA, but the mass spec data in Figure 3 implies that at least some of the population, possibly half, is un-modified tyrosine. Is it possible that the residue would be more accurately modeled with partial occupancy of both DOPA and tyrosine?

Minor note: We appreciate the addition of Supporting Figure 3. The caption can be made clearer with what model the density shown was refined against for each of the top and bottom rows.

Data analysis, methodology, interpretation, and conclusions

The authors addressed the errors and oversights we noted in the previous review. The section on operon structure now is clearer. Overall, the methodology is sound, and reproduction is possible from the methods section.

Reviewer #3

(Remarks to the Author)

1. The abstract is quite field-specific with a high level of detail. The reviewers ask the authors to re-work so the overarching impact or significance of the paper's findings is clear.

We appreciate the comment and have edited the abstract and removed some of the details.

This is ok.

2. Figure 1: The caption should explicitly state that this is a species tree, not a gene tree of RNR Class Ie sequences with specific mention that RNR annotations were plotted onto the GTDB phylogeny with a citation for the original work. The reviewers assume the tree was rooted according to the GTDB analyses; please confirm this and clarify in the caption and/or methods section.

We thank the reviewer for this comment and have edited the caption of Figure 1 accordingly.

This is ok.

3. How does the Class Ie distribution, when mapped onto the species tree, compare to known gene tree organizations of the VPK vs. QSK sequence clustering? Are there any anomalies indicative of horizontal transfer, or were all Class Ie linearly inherited?

Since the genes are encoded by distantly related genomes, HGT was very likely involved in their evolution. We have added a sentence noting this in the organism distribution paragraph.

The reviewers are confused by this statement. If the tree presented in figure 1 is indeed a species tree (as clarified above), it appears to show a clear case of linear inheritance with QSK and VPK organisms appearing only once in distinct regions of the tree. Perhaps the reviewers' comments may have been misconstrued by the authors. We were simply wondering if the authors could confirm the linear inheritance apparent in their current tree with a rudimentary gene tree.

4. Line 98-99: The authors state that the QSK variant of Class Ie is the evolutionary bridge between Class Ib and the VPK variant but provide no citation or data to support this statement. Based on the annotated species tree provided, it appears the opposite may be true, with the Fusobacteriales outgroup exclusively containing the VPK motif. Can the authors justify this statement? Additionally, the reviewers recommend including canonical Class Ib sequences to contextualize the Class Ie clades in figure 1.

Our claim that the QSK variant is the evolutionary bridge to the VPK variant was based on the phylogeny presented in Srinivas et al. 2018. To clarify this, we have added a citation to the paper in the paragraph regarding the organism distribution.

It would be highly impractical to include the distribution of class Ib in Figure 1. There are >6500 genomes encoding at least one *nrdE* or *nrdF*, distributed over 10 phyla and 16 classes in the GTDB taxonomy. The protein phylogeny in Srinivas et al. 2018 contains several class Ib *NrdF* sequences to anchor the origin of Ie in the Ib diversity.

The reference provided by the authors is acceptable. There would be no need to include a complete distribution of class Ib, it would be satisfactory for a handful of genomes that do not contain Ie *NrdF* but do contain the Ib *NrdF* (ie those species at the cusp of this bifurcation in the Srinivas et al publication) from the GTDB. This would also help establish the linear inheritance hypothesis. If the Ib sequences sit as effectively an outgroup on your species tree in addition to the position in the published gene tree in Srinivas this would be very convincing.

5. Figure 3: The reviewers note the large standard deviation of results with R2/*NrdI*/*NrdH* and R2/*R1*/*NrdI*/*NrdH* combinations in both panels of figure 2. Despite this, the authors maintain that the addition of R1 corresponds with an increase in DOPA formation. Could the authors provide a statistical assessment for significance on these samples or comment on/account for the increasing error in measurements with these protein combinations when discussing these results (paragraph at lines 203-211)?

We thank the reviewer for this valuable hint. We calculated unpaired t-test between the individual DOPA ratios and edited Figure 3B accordingly. We also added the p-values for the tests in the text and added a sentence to the MS methods.

This is OK.

6. Table 1: The dataset presented in Table 1 is borderline acceptable for the DOPA complex for data quality in the highest resolution shell, making it potentially less reliable for high-resolution details. Specifically, the low mean $I/\sigma(I)$ in the highest resolution shell indicates weak data quality, further evidenced by high R_{merge} , R_{meas} , and R_{pim} values, as well as lower $CC1/2$ and C^* values, suggesting significant noise and weaker data confidence. Likewise, the refined structure has high R_{free} in the out shell. Thus the highest resolution shell exhibits data quality issues, potentially indicating over-sampling and/or noise. Can the authors justify the use of this marginal dataset? A more reasonable cutoff ($CC1/2 > 0.5$ for e.g.) should be presented and it should be shown whether the maps are actually improved at this higher resolution.

We agree with the reviewer that we should have explained our reasoning regarding the statistics. We used the tool PAIRREF to check if our chosen high-resolution cut-off is reasonable. This tool uses the paired refinement protocol developed by Karplus & Diederichs 2012 to include all resolution shells that add improvements to the model. It showed that our highest resolution shell improves the R-values in comparison to a lower resolution cut-off at 2.0 or 2.1 Å. We added a line in the methods accordingly.

The authors just mention they used pairref in the methods but provide no details from what I can see. The data quality in outer shell is clearly very low and I see no evidence that it improves the maps or is necessary to include, but I leave it to the authors, it is a matter of preference and doesn't change much.

7: The structural characterization of the DOPA complex needs more support. The 2 sigma composite omit map looks ok at a stretch, but I would like to see a difference density omit map (3 sigma) when modelled with Tyr to demonstrate the presence of the OH group. This should be performed with the data truncated at 1.9 and approx. 2Å to confirm the additional data is useful or noise.

We appreciate the comment and agree that more detail would be beneficial here. We added Supplementary Figure 3, including the composite omit maps for residue 150 in all four protomers in the asymmetric unit as well as the difference density maps for the same residues modeled as tyrosine. The figures clearly show positive difference density in the position of the extra oxygen. However, we also want to emphasize that we used the same protein as tested by mass spectrometry and from this data we know that the protein contains a DOPA residue in this peptide, the assignment of a DOPA is thus not solely based on the structural data. We also know from mass spectrometry data that a fraction of the protein is not modified and contains a tyrosine at position 150. Thus, while we clearly see additional density compared to a tyrosine, we do not want to claim an occupancy of the additional OH group of 100%. It is however important to show the position of the additional OH group to be able to compare it to the VPK version.

The omit density is convincing, this is ok.

Reviewer #4

(Remarks to the Author)

Reviewer #5

(Remarks to the Author)

Reviewer #6

(Remarks to the Author)

I have been asked to "look at the revised manuscript to gauge the authors' rebuttal to the comments from Reviewer #1."

In the original review of this manuscript Reviewer #1 identified three problematic issues that reduced his/her enthusiasm for the work reported by John et al.

1. No mechanistic or scientific rationale was presented for the apparent increase in DOPA formation in R2 upon co-expression of NrdH, NrdI, Nrd/NrdH, or NrdI/NrdH/R1.

In the revised manuscript the authors did not provide a mechanistic or scientific rationale for the increase in DOPA formation in R2 upon the co-expression of the other genes identified in the apparent gene cluster.

2. The solved structures did not provide novel insights for the generation of the putative radical in catalysis or DOPA installation in R2eQSK relative to R2eVPK.

In the revised manuscript the authors did not identify any differences in the structure of R2eQSK relative to that of R2eVPK that would explain the functional differences between these two proteins.

3. It was unclear as to why the x-axis in Figures 2B and 3A were modified.

The authors have apparently corrected this issue.

I am in agreement with Reviewer #1 that the presentation provides only an incremental advance toward our understanding of the R2eQSK variant of RNR and that the authors did not attempt to address the points raised by Reviewer #1 regarding points #1 and #2.

Version 1:

Reviewer comments:

Reviewer #2

(Remarks to the Author)

We have read through the edits to the manuscript as well as the rebuttal. The authors have responded in fair fashion to our questions. After the further edits, we think that the manuscript reads clearly. The introduction is easier to follow. We agree with the authors that with these changes, an additional introduction figure is not needed.

The inclusion of the activity assay data (now Figure 4) in the main text is much appreciated. The reaction scheme added to the figure will help the casual reader understand what's happening in the figure. The Figure 4 title and/or caption may benefit from a little more detail or specificity.

SI Figures 2 and 3 are both clear now.

We think that this work is technically sound and of good merit. The conclusions are supported by the data. As no further experiments have been performed, the significance and noteworthiness of this work has not changed significantly since our prior reviews.

Reviewer #3

(Remarks to the Author)

Reviewer #3 (Remarks to the Author):

1. The abstract is quite field-specific with a high level of detail. The reviewers ask the authors to re-work so the overarching impact or significance of the paper's findings is clear.

We appreciate the comment and have edited the abstract and removed some of the details.

This is ok.

2. Figure 1: The caption should explicitly state that this is a species tree, not a gene tree of RNR Class Ie sequences with specific mention that RNR annotations were plotted onto the GTDB phylogeny with a citation for the original work. The reviewers assume the tree was rooted according to the GTDB analyses; please confirm this and clarify in the caption and/or methods section.

We thank the reviewer for this comment and have edited the caption of Figure 1 accordingly.

This is ok.

3. How does the Class Ie distribution, when mapped onto the species tree, compare to known gene tree organizations of the VPK vs. QSK sequence clustering? Are there any anomalies indicative of horizontal transfer, or were all Class Ie linearly inherited

Since the genes are encoded by distantly related genomes, HGT was very likely involved in their evolution. We have added a sentence noting this in the organism distribution paragraph.

The reviewers are confused by this statement. If the tree presented in figure 1 is indeed a species tree (as clarified above), it appears to show a clear case of linear inheritance with QSK and VPK organisms appearing only once in distinct regions of the tree. Perhaps the reviewers comments may have been misconstrued by the authors. We were simply wondering if the authors could confirm the linear inheritance apparent in their current tree with a rudimentary gene tree.

We published a phylogeny of the proteins in Srinivas et al. 2018, whereas in this study we chose to only show the distribution of QSK and VPK enzymes among extant bacteria. The current tree does not purport to show any detailed evolutionary processes such as linear or horizontal transfer of genes. The only claim we make regarding such processes in the current manuscript is what we added after your earlier comment:

“This distribution strongly suggests horizontal gene transfer of *nrdF* QSK and *nrdF* VPK genes between distantly related genomes, in line with what has previously been shown as a general pattern for RNR genes.”

This was based on the fact that QSK genes are found in several actinobacterial orders and that VPK genes are found in several phyla (most prominently Actinomycetota and Bacillota_1, suggesting HGT from a member of the former to a member of the latter). Complete linear inheritance would hence suggest that both QSK and VPK genes were found in the common ancestor of these phyla and massive loss in most lineages, which is clearly not likely and certainly not what the reviewers suggest. That linear transfer still is the dominating mode of inheritance among extant organisms – as is the case for virtually all genes, except possibly for “selfish” genetic elements – we believe is obvious. We made this clearer by modifying the above sentence to:

“This distribution strongly suggests an influence of horizontal gene transfer of *nrdF* QSK and *nrdF* VPK genes between distantly related genomes, in line with what has previously been shown as a general pattern for RNR genes.”

A deeper analysis of inheritance patterns of the genes would be interesting but is outside our intended scope of the current manuscript.

This is fine.

4. Line 98-99: The authors state that the QSK variant of Class Ie is the evolutionary bridge between Class Ib and the VPK variant but provide no citation or data to support this statement. Based on the annotated species tree provided, it appears the opposite may be true, with the Fusobacteriales outgroup exclusively containing the VPK motif. Can the authors justify this statement? Additionally, the reviewers recommend including canonical Class Ib sequences to contextualize the Class Ie clades in figure 1.

Our claim that the QSK variant is the evolutionary bridge to the VPK variant was based on the phylogeny presented in Srinivas et al. 2018. To clarify this, we have added a citation to the paper in the paragraph regarding the organism distribution.

It would be highly impractical to include the distribution of class Ib in Figure 1. There are >6500 genomes encoding at least one *nrdE* or *nrdF*, distributed over 10 phyla and 16 classes in the GTDB taxonomy. The protein phylogeny in Srinivas et al. 2018 contains several class Ib *NrdF* sequences to anchor the origin of Ie in the Ib diversity.

The reference provided by the authors is acceptable. There would be no need to include a complete distribution of class Ib, it would be satisfactory for a handful of genomes that do not contain Ie *NrdF* but do contain the Ib *NrdF* (ie those species at the cusp of this bifurcation in the Srinivas et al publication) from the GTDB. This would also help establish the linear inheritance hypothesis. If the Ib sequences sit as effectively an outgroup on your species tree in addition to the position in the published gene tree in Srinivas this would be very convincing.

Reviewer #4

(Remarks to the Author)

Reviewer #3 (Remarks to the Author):

1. The abstract is quite field-specific with a high level of detail. The reviewers ask the authors to re-work so the overarching impact or significance of the paper's findings is clear.

We appreciate the comment and have edited the abstract and removed some of the details.

This is ok.

2. Figure 1: The caption should explicitly state that this is a species tree, not a gene tree of RNR Class Ie sequences with specific mention that RNR annotations were plotted onto the GTDB phylogeny with a citation for the original work. The reviewers assume the tree was rooted according to the GTDB analyses; please confirm this and clarify in the caption and/or methods section.

We thank the reviewer for this comment and have edited the caption of Figure 1 accordingly.

This is ok.

3. How does the Class Ie distribution, when mapped onto the species tree, compare to known gene tree organizations of the VPK vs. QSK sequence clustering? Are there any anomalies indicative of horizontal transfer, or were all Class Ie linearly inherited

Since the genes are encoded by distantly related genomes, HGT was very likely involved in their evolution. We have added a sentence noting this in the organism distribution paragraph.

The reviewers are confused by this statement. If the tree presented in figure 1 is indeed a species tree (as clarified above), it appears to show a clear case of linear inheritance with QSK and VPK organisms appearing only once in distinct regions of the tree. Perhaps the reviewers comments may have been misconstrued by the authors. We were simply wondering if the authors could confirm the linear inheritance apparent in their current tree with a rudimentary gene tree.

We published a phylogeny of the proteins in Srinivas et al. 2018, whereas in this study we chose to only show the

distribution of QSK and VPK enzymes among extant bacteria. The current tree does not purport to show any detailed evolutionary processes such as linear or horizontal transfer of genes. The only claim we make regarding such processes in the current manuscript is what we added after your earlier comment:

“This distribution strongly suggests horizontal gene transfer of *nrdF* QSK and *nrdF* VPK genes between distantly related genomes, in line with what has previously been shown as a general pattern for RNR genes.”

This was based on the fact that QSK genes are found in several actinobacterial orders and that VPK genes are found in several phyla (most prominently Actinomycetota and Bacillota_I, suggesting HGT from a member of the former to a member of the latter). Complete linear inheritance would hence suggest that both QSK and VPK genes were found in the common ancestor of these phyla and massive loss in most lineages, which is clearly not likely and certainly not what the reviewers suggest. That linear transfer still is the dominating mode of inheritance among extant organisms – as is the case for virtually all genes, except possibly for “selfish” genetic elements – we believe is obvious. We made this clearer by modifying the above sentence to:

“This distribution strongly suggests an influence of horizontal gene transfer of *nrdF* QSK and *nrdF* VPK genes between distantly related genomes, in line with what has previously been shown as a general pattern for RNR genes.”

A deeper analysis of inheritance patterns of the genes would be interesting but is outside our intended scope of the current manuscript.

This is fine.

4. Line 98-99: The authors state that the QSK variant of Class Ie is the evolutionary bridge between Class Ib and the VPK variant but provide no citation or data to support this statement. Based on the annotated species tree provided, it appears the opposite may be true, with the Fusobacteriales outgroup exclusively containing the VPK motif. Can the authors justify this statement? Additionally, the reviewers recommend including canonical Class Ib sequences to contextualize the Class Ie clades in figure 1.

Our claim that the QSK variant is the evolutionary bridge to the VPK variant was based on the phylogeny presented in Srinivas et al. 2018. To clarify this, we have added a citation to the paper in the paragraph regarding the organism distribution.

It would be highly impractical to include the distribution of class Ib in Figure 1. There are >6500 genomes encoding at least one *nrdE* or *nrdF*, distributed over 10 phyla and 16 classes in the GTDB taxonomy. The protein phylogeny in Srinivas et al. 2018 contains several class Ib *NrdF* sequences to anchor the origin of Ie in the Ib diversity.

The reference provided by the authors is acceptable. There would be no need to include a complete distribution of class Ib, it would be satisfactory for a handful of genomes that do not contain Ie *NrdF* but do contain the Ib *NrdF* (ie those species at the cusp of this bifurcation in the Srinivas et al publication) from the GTDB. This would also help establish the linear inheritance hypothesis. If the Ib sequences sit as effectively an outgroup on your species tree in addition to the position in the published gene tree in Srinivas this would be very convincing.

Our claim that QSK Ie is the evolutionary bridge between class Ib and Ie – i.e. that QSK is ancestral to VPK and that the former evolved directly from a Ib enzyme – is not dependent on the extant distribution of neither Ib nor Ie genes – which is why we do not refer to figure 1 of the current manuscript in support of this – but is clearly shown by the phylogeny in Srinivas et al. 2018.

Regarding the linear or horizontal transfer of genes, see our reply above.

This is ok. While we maintain it would add to the clarity of the piece, the reviewers are satisfied it is sufficient to provide the reference.

5. Figure 3: The reviewers note the large standard deviation of results with R2/*NrdI*/*NrdH* and R2/R1/*NrdI*/*NrdH* combinations in both panels of figure 2. Despite this, the authors maintain that the addition of R1 corresponds with an increase in DOPA formation. Could the authors provide a statistical assessment for significance on these samples or comment on/account for the increasing error in measurements with these protein combinations when discussing these results (paragraph at lines 203-211)?

We thank the reviewer for this valuable hint. We calculated unpaired t-test between the individual DOPA ratios and edited Figure 3B accordingly. We also added the p-values for the tests in the text and added a sentence to the MS methods.

This is OK.

6. Table 1: The dataset presented in Table 1 is borderline acceptable for the DOPA complex for data quality in the highest resolution shell, making it potentially less reliable for high-resolution details. Specifically, the low mean $1/\sigma(I)$ in the highest resolution shell indicates weak data quality, further evidenced by high *R_{merge}*, *R_{meas}*, and *R_{pim}* values, as well as lower *CC_{1/2}* and *C** values, suggesting significant noise and weaker data confidence. Likewise, the refined structure has high *R_{free}* in the out shell. Thus the highest resolution shell exhibits data quality issues, potentially indicating over-sampling and/or noise. Can the authors justify the use of this marginal dataset? A more reasonable cutoff (*CC_{1/2}* >0.5 for e.g.) should be presented and it should be shown whether the maps are actually improved at this higher resolution.

We agree with the reviewer that we should have explained our reasoning regarding the statistics. We used the tool PAIRREF to check if our chosen high-resolution cut-off is reasonable. This tool uses the paired refinement protocol developed by Karplus & Diederichs 2012 to include all resolution shells that add improvements to the model. It showed that our highest resolution shell improves the R-values in comparison to a lower resolution cut-off at 2.0 or 2.1 Å. We added a line in the methods accordingly.

The authors just mention they used pairref in the methods but provide no details from what I can see. The data quality in outer shell is clearly very low and I see no evidence that it improves the maps or is necessary to include, but I leave it to the authors, it is a matter of preference and doesn't change much.

We added details of the input and output of PAIREF.

This is ok.

7: The structural characterization of the DOPA complex needs more support. The 2 sigma composite omit map looks ok at a stretch, but I would like to see a difference density omit map (3 sigma) when modelled with Tyr to demonstrate the presence of the OH group. This should be performed with the data truncated at 1.9 and approx. 2Å to confirm the additional data is useful or noise.

We appreciate the comment and agree that more detail would be beneficial here. We added Supplementary Figure 3, including the composite omit maps for residue 150 in all four protomers in the asymmetric unit as well as the difference density maps for the same residues modeled as tyrosine. The figures clearly show positive difference density in the position of the extra oxygen. However, we also want to emphasize that we used the same protein as tested by mass spectrometry and from this data we know that the protein contains a DOPA residue in this peptide, the assignment of a DOPA is thus not solely based on the structural data. We also know from mass spectrometry data that a fraction of the protein is not modified and contains a tyrosine at position 150. Thus, while we clearly see additional density compared to a tyrosine, we do not want to claim an occupancy of the additional OH group of 100%. It is however important to show the position of the additional OH group to be able to compare it to the VPK version.

The omit density is convincing, this is ok.

Reviewer #5

(Remarks to the Author)

Reviewer #6

(Remarks to the Author)

Regarding the comments of the original reviewer #1, the authors have now addressed the two comments that were ignored in the first revision of this manuscript. They have now admitted that they do not understand the increase in DOPA formation in R2 upon co-expression of the additional proteins. This is fine with me. Secondly, they have now added Supplementary Figure 3 which compares the structures of R2eQSK and R2eVPK with one another in an attempt to understand the functional differences between these two proteins. The structures are indeed quite similar to one another and they now suggest that perhaps the hydrogen bonding network is sufficiently different to facilitate differences in the catalytic properties. The authors have now adequately addressed the concerns of Reviewer #1

Reviewers' comments:

Reviewer #2 (Remarks to the Author):

Noteworthy results

In our previous review of this work, we noted that while most of the experiments performed were sound, we hoped for a more in-depth characterization of the class Ie QSK RNR to increase the overall noteworthiness of the manuscript. The authors have addressed our questions of interpretation, fixed minor issues or errors, and explained in their responses to the first review why the further experiments we suggested might be challenging. Because they did not perform further experiments, the significance of the results has not increased substantially. As this manuscript is now submitted to Communications Biology, we have tried to compare to other papers published in this journal. We feel that the limited quantitative results of the enzymology in this work is a little below the norm for this journal, but may be acceptable due to the challenges of developing a quantitative assay for the enzyme of interest.

Significance of work

This work is of good merit, and as noted in our previous review, it is of significance to the field of enzymology and RNR diversity. The authors have updated text in the manuscript to more clearly define what was previously known in Srinivas et al. 2018 and how this paper is building upon this. The introduction is still a bit confusing to people unfamiliar with RNRs. One potential suggestion: a figure in the introduction that visually communicates the mechanism and purpose of each subunit would benefit the reader, as the written description in lines 39 to 56 is verbose.

We feel that a figure of the general mechanism and working of different types of RNRs would draw attention from the message of the paper. The reviews cited contain good overviews including great figures for the interested reader. We simplified the mentioned part of the introduction, removing some details about other class I RNRs not relevant for this study, thus making it easier to follow.

Does the work support the conclusions and claims?

The authors have addressed most of our questions for the following two points:

1. We asked for further activity characterization and more compelling evidence of the DOPA radical. The authors addressed our questions in their "response to reviewers" and explained why quantitative activity assays and EPR or UV/vis experiments would be challenging under current circumstances. This is understandable, but it does make for a less compelling manuscript.

Fundamentally, we are curious about the cause of what seems to be overall low activity of the protein of interest as compared to prior RNRs, including that characterized by Srinivas et al. We would like to confirm that Figure 2c iron content is in fact reported in %, as written (e.g., 0.33%), and not in mol/mol fraction (e.g., 0.33 or 33%). Were it the latter, we would wonder if the RNR is in fact a metalloprotein with poor holo-protein content.

The molarity of the measured protein samples lies between 154 and 575 μM and the measured iron content between 0.466 and 0.842 μM as mentioned in Supplementary table 3, so the number in the main text are correct (0.33%) and there is thus very little iron in the measured R2 protein.

Minor note: We feel that inclusion of the activity assay results (Supplemental Figure 1) in the main text would be relevant to show the reader that the purified proteins are active for ribonucleotide reduction. The figure could be paired with a scheme of the chemical reaction, which would help the reader follow the section “In vitro activity of BvRNRQSK” (lines 225 to 244). The noteworthy conclusion that the DOPA modification and NrdI are required for RNR activity is substantiated by the data, and therefore it would benefit the reader to see the data in the main text.

We agree with the reviewer that having the results included in the main text might benefit the understanding while reading through the paragraph. We have modified supplementary figure 1 to include a simplified reaction scheme and moved it to the end of the paragraph “In vitro activity of BvRNR” as a new Figure 4.

2. We posed questions about the data quality of the 1.9 Å crystal structure with a DOPA radical. The authors have responded with a reasonable explanation for their resolution cut-off that was used. We do still wonder about the occupancy of the DOPA. The text describes it as clearly DOPA, but the mass spec data in Figure 3 implies that at least some of the population, possibly half, is un-modified tyrosine. Is it possible that the residue would be more accurately modeled with partial occupancy of both DOPA and tyrosine?

It is possible that a fraction of the protein in the structure contains tyrosine instead of DOPA. From a technical perspective, it would be more precise to figure out the ratio of DOPA to tyr in the crystal. However, the mass spec data does not determine absolute concentrations of DOPA and does not necessarily reflect the situation in the crystal. In principle, it would be possible to refine the occupancy of the additional oxygen atom in DOPA, but at the current resolution, it would yield imprecise results. As the Fo-Fc map looks clean after modelling it with a DOPA, we decided to model the DOPA with full occupancy.

Minor note: We appreciate the addition of Supporting Figure 3. The caption can be made clearer with what model the density shown was refined against for each of the top and bottom rows.

We have edited the caption of this Supplementary Figure accordingly (now Supplementary Figure 2).

Data analysis, methodology, interpretation, and conclusions

The authors addressed the errors and oversights we noted in the previous review. The section on operon structure now is clearer. Overall, the methodology is sound, and reproduction is possible from the methods section.

Reviewer #3 (Remarks to the Author):

1. The abstract is quite field-specific with a high level of detail. The reviewers ask the authors to re-work so the overarching impact or significance of the paper's findings is clear.

We appreciate the comment and have edited the abstract and removed some of the details.

This is ok.

2. Figure 1: The caption should explicitly state that this is a species tree, not a gene tree of RNR Class Ie sequences with specific mention that RNR annotations were plotted onto the GTDB phylogeny with a citation for the original work. The reviewers assume the tree was rooted according to the GTDB analyses; please confirm this and clarify in the caption and/or methods section.

We thank the reviewer for this comment and have edited the caption of Figure 1 accordingly.

This is ok.

3. How does the Class Ie distribution, when mapped onto the species tree, compare to known gene tree organizations of the VPK vs. QSK sequence clustering? Are there any anomalies indicative of horizontal transfer, or were all Class Ie linearly inherited

Since the genes are encoded by distantly related genomes, HGT was very likely involved in their evolution. We have added a sentence noting this in the organism distribution paragraph.

The reviewers are confused by this statement. If the tree presented in figure 1 is indeed a species tree (as clarified above), it appears to show a clear case of linear inheritance with QSK and VPK organisms appearing only once in distinct regions of the tree. Perhaps the reviewers comments may have been misconstrued by the authors. We were simply wondering if the authors could confirm the linear inheritance apparent in their current tree with a rudimentary gene tree.

We published a phylogeny of the proteins in Srinivas et al. 2018, whereas in this study we chose to only show the distribution of QSK and VPK enzymes among extant bacteria. The current tree does not purport to show any detailed evolutionary processes such as linear or horizontal transfer of genes. The only claim we make regarding such processes in the current manuscript is what we added after your earlier comment:

“This distribution strongly suggests horizontal gene transfer of *nrdF* QSK and *nrdF* VPK genes between distantly related genomes, in line with what has previously been shown as a general pattern for RNR genes.”

This was based on the fact that QSK genes are found in several actinobacterial *orders* and that VPK genes are found in several *phyla* (most prominently *Actinomycetota* and *Bacillota_I*, suggesting HGT from a member of the former to a member of the latter).

Complete linear inheritance would hence suggest that both QSK and VPK genes were found in the common ancestor of these phyla and massive loss in most lineages, which is clearly not likely and certainly not what the reviewers suggest. That linear transfer still is the dominating mode of inheritance among extant organisms – as is the case for virtually all genes, except possibly for “selfish” genetic elements – we believe is obvious. We made this clearer by modifying the above sentence to:

“This distribution strongly suggests an influence of horizontal gene transfer of *nrdF* QSK and *nrdF* VPK genes between distantly related genomes, in line with what has previously been shown as a general pattern for RNR genes.”

A deeper analysis of inheritance patterns of the genes would be interesting but is outside our intended scope of the current manuscript.

4. Line 98-99: The authors state that the QSK variant of Class Ie is the evolutionary bridge between Class Ib and the VPK variant but provide no citation or data to support this statement. Based on the annotated species tree provided, it appears the opposite may be true, with the Fusobacteriales outgroup exclusively containing the VPK motif. Can the authors justify this statement? Additionally, the reviewers recommend including canonical Class Ib sequences to contextualize the Class Ie clades in figure 1.

Our claim that the QSK variant is the evolutionary bridge to the VPK variant was based on the phylogeny presented in Srinivas et al. 2018. To clarify this, we have added a citation to the paper in the paragraph regarding the organism distribution.

It would be highly impractical to include the distribution of class Ib in Figure 1. There are >6500 genomes encoding at least one *nrdE* or *nrdF*, distributed over 10 phyla and 16 classes in the GTDB taxonomy. The protein phylogeny in Srinivas et al. 2018 contains several class Ib *NrdF* sequences to anchor the origin of Ie in the Ib diversity.

The reference provided by the authors is acceptable. There would be no need to include a complete distribution of class Ib, it would be satisfactory for a handful of genomes that do not contain Ie *NrdF* but do contain the Ib *NrdF* (ie those species at the cusp of this bifurcation in the Srinivas et al publication) from the GTDB. This would also help establish the linear inheritance hypothesis. If the Ib sequences sit as effectively an outgroup on your species tree in addition to the position in the published gene tree in Srinivas this would be very convincing.

Our claim that QSK Ie is the evolutionary bridge between class Ib and Ie – i.e. that QSK is ancestral to VPK and that the former evolved directly from a Ib enzyme – is not dependent on the extant distribution of neither Ib nor Ie genes – which is why we do not refer to figure 1 of the current manuscript in support of this – but is clearly shown by the phylogeny in Srinivas et al. 2018.

Regarding the linear or horizontal transfer of genes, see our reply above.

5. Figure 3: The reviewers note the large standard deviation of results with R2/*NrdI*/*NrdH* and R2/R1/*NrdI*/*NrdH* combinations in both panels of figure 2. Despite this, the authors maintain that the addition of R1 corresponds with an increase in

DOPA formation. Could the authors provide a statistical assessment for significance on these samples or comment on/account for the increasing error in measurements with these protein combinations when discussing these results (paragraph at lines 203-211)?

We thank the reviewer for this valuable hint. We calculated unpaired t-test between the individual DOPA ratios and edited Figure 3B accordingly. We also added the p-values for the tests in the text and added a sentence to the MS methods.

This is OK.

6. Table 1: The dataset presented in Table 1 is borderline acceptable for the DOPA complex for data quality in the highest resolution shell, making it potentially less reliable for high-resolution details. Specifically, the low mean $I/\sigma(I)$ in the highest resolution shell indicates weak data quality, further evidenced by high R_{merge} , R_{meas} , and R_{pim} values, as well as lower $CC1/2$ and C^* values, suggesting significant noise and weaker data confidence. Likewise, the refined structure has high R_{free} in the out shell. Thus the highest resolution shell exhibits data quality issues, potentially indicating over-sampling and/or noise. Can the authors justify the use of this marginal dataset? A more reasonable cutoff ($CC1/2 > 0.5$ for e.g.) should be presented and it should be shown whether the maps are actually improved at this higher resolution.

We agree with the reviewer that we should have explained our reasoning regarding the statistics. We used the tool PAIREF to check if our chosen high-resolution cut-off is reasonable. This tool uses the paired refinement protocol developed by Karplus & Diederichs 2012 to include all resolution shells that add improvements to the model. It showed that our highest resolution shell improves the R-values in comparison to a lower resolution cut-off at 2.0 or 2.1 Å. We added a line in the methods accordingly.

The authors just mention they used pairef in the methods but provide no details from what I can see. The data quality in outer shell is clearly very low and I see no evidence that it improves the maps or is necessary to include, but I leave it to the authors, it is a matter of preference and doesn't change much.

We added details of the input and output of PAIREF.

7: The structural characterization of the DOPA complex needs more support. The 2 sigma composite omit map looks ok at a stretch, but I would like to see a difference density omit map (3 sigma) when modelled with Tyr to demonstrate the presence of the OH group. This should be performed with the data truncated at 1.9 and approx. 2Å to confirm the additional data is useful or noise.

We appreciate the comment and agree that more detail would be beneficial here. We added Supplementary Figure 3, including the composite omit maps for residue 150 in all four protomers in the asymmetric unit as well as the difference density maps for the same residues modeled as tyrosine. The figures clearly show positive difference density in the position of the extra oxygen. However, we also want to emphasize that we used the same protein as tested by mass spectrometry and from this data we know that the protein contains a DOPA residue in this peptide, the assignment of a DOPA is thus not solely based on the structural data. We also know from mass spectrometry

data that a fraction of the protein is not modified and contains a tyrosine at position 150. Thus, while we clearly see additional density compared to a tyrosine, we do not want to claim an occupancy of the additional OH group of 100%. It is however important to show the position of the additional OH group to be able to compare it to the VPK version.

The omit density is convincing, this is ok.

Reviewer #6 (Remarks to the Author):

I have been asked to “look at the revised manuscript to gauge the authors' rebuttal to the comments from Reviewer #1.”

In the original review of this manuscript Reviewer #1 identified three problematic issues that reduced his/her enthusiasm for the work reported by John et al.

1. No mechanistic or scientific rationale was presented for the apparent increase in DOPA formation in R2 upon co-expression of NrdH, NrdI, Nrd/NrdH, or NrdI/NrdH/R1.

In the revised manuscript the authors did not provide a mechanistic or scientific rationale for the increase in DOPA formation in R2 upon the co-expression of the other genes identified in the apparent gene cluster.

This observation is correct. The reason why we chose not to discuss mechanistic rationales is that the data this study is based on does not provide information to suggest a mechanistic model. Any mechanistic discussion would be pure speculation and we prefer to avoid that. We agree with the reviewer that the mechanism behind our observation is of high interest and is the topic of ongoing studies in our lab.

2. The solved structures did not provide novel insights for the generation of the putative radical in catalysis or DOPA installation in R2eQSK relative to R2eVPK.

In the revised manuscript the authors did not identify any differences in the structure of R2eQSK relative to that of R2eVPK that would explain the functional differences between these two proteins.

We have taken this remark into consideration and we have now revisited our structural comparison. A more zoomed-out view of the active site shows that the water network between the two proteins is different, which likely leads to different behaviour in the context of shuttling the oxidant to the DOPA and generating and stabilising the radical. We have added the Supplementary Figure 3, included a paragraph at the end of the section “Structures of BvR2”, and edited the section regarding the structures in the discussion.

3. It was unclear as to why the x-axis in Figures 2B and 3A were modified.

The authors have apparently corrected this issue.

I am in agreement with Reviewer #1 that the presentation provides only an incremental advance toward our understanding of the R2eQSK variant of RNR and that the authors did not attempt to address the points raised by Reviewer #1 regarding points #1 and #2.